# Engineered Tissue for Cardiac Regeneration: Current Status and Future Perspectives

**DOI:** 10.3390/bioengineering9110605

**Published:** 2022-10-22

**Authors:** Junjun Li, Li Liu, Jingbo Zhang, Xiang Qu, Takuji Kawamura, Shigeru Miyagawa, Yoshiki Sawa

**Affiliations:** 1Department of Cardiovascular Surgery, Osaka University Graduate School of Medicine, Suita, Osaka 565-0871, Japan; 2Cardiovascular Division, Osaka Police Hospital, Tennoji, Osaka 543-0035, Japan

**Keywords:** heart failure, human induced pluripotent stem cells, cardiac regeneration, tissue engineering, cardiomyocytes

## Abstract

Heart failure (HF) is the leading cause of death worldwide. The most effective HF treatment is heart transplantation, the use of which is restricted by the limited supply of donor hearts. The human pluripotent stem cell (hPSC), including human embryonic stem cell (hESC) and the induced pluripotent stem cells (hiPSC), could be produced in an infinite manner and differentiated into cardiomyocytes (CMs) with high efficiency. The hPSC-CMs have, thus, offered a promising alternative for heart transplant. In this review, we introduce the tissue-engineering technologies for hPSC-CM, including the materials for cell culture and tissue formation, and the delivery means into the heart. The most recent progress in clinical application of hPSC-CMs is also introduced. In addition, the bottleneck limitations and future perspectives for clinical translation are further discussed.

## 1. Introduction

Heart failure (HF) refers to a condition in which the heart is unable to pump blood into the body. It is a major cause of death in both developed and developing countries [1]. With over 30 million patients worldwide, HF has been listed as a crucial public healthcare concern, owing to its high prevalence, mortality, morbidity, and cost of care [2]. Reduced cardiac contractility and function, irregular left ventricle remodeling, and uneven stress distribution in the heart muscle, which occur after myocardial infarction (MI), eventually lead to catastrophic HF. Researchers and clinicians worldwide are working extensively to develop new treatment strategies to improve the survival rate and reduce the global incidence of HF.

The human heart is a complex organ that is composed of various cell types, such as cardiomyocytes (CMs), fibroblasts, and endothelial cells, which can maintain blood flow throughout the body. CMs, which make up approximately 1/3 of the cells in the atria and approximately half of the cells in the ventricles, are the smallest contractile units of the heart [3]. However, the adult human heart lacks endogenous regeneration potential; thus, any defect in the size and deficiency of CMs leads to severe MI-related cardiovascular complications, eventually resulting in HF [4].

Current HF treatments include prevention and end-of-life care, while surgical heart transplantation is the only curative remedy. However, the limited supply of donor hearts severely restricts the application of heart transplantation [5]. Moreover, immunosuppressants, which are required following allogeneic transplantation, may cause serious kidney damage or other problems, such as cancers and infections [6]. Owing to these problems, the emergence of cardiac tissue engineering has provided substantial hope not only for resolving or rescuing damaged hearts post-MI but also for providing a permanent treatment strategy. In addition to its application in cardiac regenerative medicine, cardiac tissue construction in vitro can be applied for drug toxicity or drug screening evaluation and in vitro disease modeling [3]. However, the cells and methods that should be used to construct cardiac tissue need to be further explored.

Regarding the provision of cells for tissue construction, pluripotent stem cells (PSCs) (e.g., embryonic PSCs) are a promising cell source because they can differentiate into various cells, including CMs. The embryonic PSCs, however, still have the ethical issues [7]. In 2006, Professor Shinya Yamanaka succeeded in establishing induced PSCs (iPSCs) from somatic cells, using four transcription factors [8]. Later, in order to reduce the risk of tumorigenesis, hiPSCs were generated and validated with three transcription factors [6,9]. In addition, induction method by using combinations of plasmids has been developed to efficiently induce clinical-grade hiPSCs [10].

Overall, iPSCs are attracting attention as a tool for drug discovery, regenerative medicine, and disease pathology elucidation. In particular, since the adult CMs have a low cell proliferation capacity, there are great expectations for the application of human iPSC(hiPSC)-derived CMs in regenerative medicine. However, there have been concerns regarding the safety of hiPS cells used in regenerative medicine. In recent years, a highly safe and non-invasive method producing hiPSC lines has been established, and some of these cell lines have already been clinically applied [11,12]. On the other hand, methods for producing hiPSC-derived CMs have been reported one after another [13]. Although the problem of CMs purity and quality has been largely resolved, the number of cells required for myocardial transplantation is on the order of 100 million. Therefore, various problems exist, such as the provision of large amounts of cells at low cost, CMs purification, removal of undifferentiated cells, avoidance of carcinogenic risk, and immunosuppressive effects, all of which being still refined.

In this review, we focus on the most recent techniques and progress in cardiac tissue engineering and highlight, in particular, effective therapeutic approaches for cardiac regeneration. There are two main conventional transplantation methods for treating heart failure: patch-based therapy and injection-based therapy. The overall representation and features of the aforementioned approaches are presented in Figure 1.

## 2. Engineered Cardiac Patch (ECP) with Scaffold

In 1997, Eschenhagen et al. developed the earliest version of an ECP. They used chicken embryonic CMs and collagen matrix for ECP formation [14]. In general, ECP can be constructed by combining different types of cells, bioactive molecules (growth factors), and biomaterials. The generation of ECP has improved our understanding of cardiac function both in normal and pathological conditions. ECP has been widely used in drug screening, disease modeling, and regenerative medicine [15]. A state-of-the-art work by Zhao et al. has reported a scalable ECP that enables assessment of atrial and ventricular tissue function; they electrically paced the tissue for 8 months for modeling polygenic hypertrophy [16]. Saleem et al. performed a blinded multicenter evaluation of drug-induced change in ECP contractility, which aligned closely to the free therapeutic plasma concentration, showing potential for a routine tool in safety pharmacology [17]. When used for transplantation, ECP shows a higher survival rate of engraftment and significant improvement in cardiac regeneration compared with suspended single cells [18].

### 2.1. Natural-Material-Scaffold-Based Approach

Natural material scaffolds have been widely used in ECP construction because of their low immune response, which, when aggravated, can lead to the destruction of the material [19]. Hydrogels are the most widely used biomaterials for engineering heat tissue. Many natural polymers have been used to make hydrogel for tissue culture: collagen I, the most abundant structural protein in extracellular matrix; fibrin, an insoluble protein that is product of bleeding and also main component of blood clot; alginate, a natural polysaccharide derived from brown algae which is ideal for encapsulating cells; gelatin, a denatured form of collagen I which is used to coating surface of substrate for improving cell adhesion; and Matrigel, a solubilized basement membrane matrix extracted from Engelbreth–Holm–Swarm mouse sarcoma tumors, containing primarily laminin, collagen IV, and various factors [20,21]. In addition, collagen V, a component of Matrigel that is minimally expressed in the uninjured heart and a minor component of scar tissue, has been reported to limit scar size after ischemic cardiac injury [22]. Tissue engineering with proper collagen V proportion might also be promising in the foreseeable future.

Roche et al. reported a 47- and 59-fold increase in cell retention, using human mesenchymal stem cells (MSCs) with collagen and alginate patches compared with that of suspended single cells in saline [23]. Furthermore, a wide variety of scaffolds with or without bioactive molecules have been used for cardiac regeneration. Wu et al. developed a wet adhesive hydrogel cardiac patch loaded with antioxidative, autophagy-regulating molecule capsules, and MSCs [24]. The scaffold could improve the cardiac microenvironment and enhance the survival of transplanted stem cells by scavenging reactive oxygen species and upregulating autophagy, thus facilitating angiogenesis and reducing cardiac fibrosis to efficiently repair the infarcted myocardium [24]. Wang et al. generated an ECP, using iPSC-derived CMs and decellularized natural heart ECM as scaffolds, which showed normal contractile and electrical physiology in vitro and improved cardiac function in a rat model of acute MI [25].

After their first development of the ECP [26], the Eschenhagen group further applied the human engineered heart tissue to repair the injured heart. They created engineered heart tissues by mixing hiPSCs with fibrin and transplanted these tissues into guinea pig hearts after cryoinjury. Compared with the human endothelial cell patches or cell-free patches, the hiPSC-derived ECP showed enhanced remuscularization in the infarct area, which also showed signs of CM proliferation and vascularization. Importantly, electrical coupling could be observed between the transplanted tissue and the host heart [27]. Several groups are working on generating larger three-dimensional cardiac tissues that are suitable for large animals or in clinical applications. Keller et al. developed a 30 mm × 30 mm cardiac tissue with multiple cell types, including CMs, endothelial cells, and vascular mural cells, which were mixed with Matrigel and collagen solution. The patch survived in vivo and improved cardiac function after MI [28]. Zhang et al. generated human cardiac muscle patches of clinically relevant dimensions (4 cm × 2 cm × 1.25 mm) by suspending CMs, smooth muscle cells, and endothelial cells differentiated from hiPSCs in fibrin. The engineered muscle patches considerably reduced the infarct size and improved the heart function. Importantly, no arrhythmia events were observed post transplantation [18]. More recently, the Eschenhagen group found a positive correlation between the dosage of transplanted CMs and the therapeutic effect [29]. Currently, one ECP-based clinical trial led by Dr. Philippe Menasché has been completed, and all but one patient receiving the cardiac progenitor cell-derived ECP showed an uneventful recovery [30]. An additional trial led by Dr. Wolfram-Hubertus Zimmermann was initiated in 2020 with the aim of recruiting 53 participants [31].

As a newly emerging technology, 3D bioprinting has also been used in creating thick tissues [32,33]. Cyfuse Biomedical has developed a sphere-based 3D-bioprinting technology. By using a needle array, the preprepared spheroids were organized into a patch with designed shape which demonstrated engraftment with vascularization in native rat myocardium [34]. Instead of using the preprepared tissue “brick”, scholars also tried to print the patch with bioink composed of single cells and biomaterials, such as alginate, gelatin, and fibrin [35,36]. The transplanted CMs and the HUVECs were encapsulated within hydrogel containing alginate and-fibrinogen and showed high orientation and well integration with the vasculature of the host [37]. In addition to the patch, some groups also 3D printed the heart with chambers that could pump blood [38,39]. The pumping function is, however, still limited, and these 3D-printed hearts still lack functional vasculature.

### 2.2. Synthetic-Material-Scaffold-Based Approach

Although there are many advantages to using natural material scaffolds, a serious limitation of this method is that most natural materials are derived from animals. The variation between batches of natural materials reduces their reliability [40]. Therefore, researchers have attempted to use synthetic polymer materials as alternatives to construct cardiac tissue. Polydimethylsiloxane (PDMS) is a silicone elastomer that has long been used for cell culture, owing to its numerous advantages, such as optic transparency, non-toxicity, and chemical compatibility. However, PDMS does not degrade in vivo, limiting its application in transplantation. Other biodegradable polymers have been utilized for cell culture and transplantation, including polycaprolactone (PCL), poly(L-lactide) (PLA), poly(glycolide) (PGA), and their copolymers (PLGA). Notably, PLGA has been approved by the FDA for medical application. PLA, PGA, and PLGA are all stiff and incompliant, limiting their application in soft-tissue engineering. Richard T. Tran et al. developed a novel biodegradable polymer based on citric acid, maleic anhydride, and 1,8-octanediol, referred to as poly(octamethylene maleate (anhydride) citrate) (POMaC). POMaC displayed an initial modulus between 0.03 and 1.54 MPa and was able to elongate as much as 534%.

Dhahri et al. introduced a soft Polydimethylsiloxane (PDMS) substrate to promote hPSC-CMs maturation. After transplanted into guinea pig heart, these matured hPSC-CMs showed better in vivo structure and alignment. Animals receiving matured CMs experienced improved contractile function recovery compared with that of the control group [41]. Other groups have integrated the micropattern into the substrate to guide the orientation of cardiac cells, leading to anisotropic organization of the engineered tissue and/or improved calcium-handling properties [42,43].

In addition to the substrate, electrical stimulation has also been used for pacing and maturing hPSC-CMs [44,45]. Notably, the stimulation in early differentiation stage promoted the maturation to adult-like level [44]. It could be expected that these adult-like CMs are capable of enabling the remuscularization in the MI area and lead to better functional recovery. More efforts have been devoted into fabricating patch-like scaffold in engineering CMs tissue. Kai et al. developed elastic, biodegradable poly(ε-caprolactone) (PCL) nanofiber cardiac patches loaded with MSCs, which were demonstrated to provide sufficient mechanical support, induce angiogenesis, and accelerate cardiac regeneration in a rat model of myocardial infarction [46]. Poly(lactic-co-glycolic acid) (PLGA) is another synthetic polymeric material that has been approved by the Food and Drug Administration (FDA) [47] and can be used as a scaffold for ECP construction. Our group obtained a high-quality cardiac-tissue-like construct by seeding iPSC-CMs on aligned PLGA nanofibers. Multilayered elongated CMs with alignment structure within the patches showed upregulation of cardiac markers, enhancement of extracellular recording, and robust drug response [48] (Figure 2). More recently, we developed large-scale functional cardiac patches by using PLGA scaffolds and iPSC-CMs and showed marked improvement in cardiac function with angiogenesis and antifibrotic effects in a porcine cardiomyopathy model [49]. In addition, our group engineered a three-dimensional cardiac patch in a rotating wall vessel bioreactor; in this way, more mature CMs were obtained and transplanted into rats with MI [50]. Lancaster et al. developed a tissue patch composed of a bioabsorbable knitted mesh, human neonatal fibroblasts, and hiPSC-CMs. The patch increased cytokine expression, enhanced electrical conduction, and improved heart function in a rat HF model [51]. Most of the present patch-based therapies still require opening the chest to deliver and mount the patch, which causes possible extra damage to the recipient. The Radisic group has developed a flexible shape-memory scaffold for minimally invasive delivery of functional tissues. As a proof of concept, they achieved successful delivery of the cardiac patches through an orifice as small as 1 mm [52]. Masumoto et al. developed a device for minimally invasive delivery of cell sheets without scaffolds. The sheets were successfully deployed within 3 min [53]. Because some ventricular arrhythmias are caused by scarring in the heart that could alter the overall conductive properties, noncellular conductive scaffolds have also been developed to restore conduction and reduce the risk of reentrant ventricular arrhythmias [54].

The oxygen diffusion limit could lead to the apoptosis of the cells within thick tissue. Many efforts were devoted to fabricating the functional vascular network. The Okano group developed a perfusion bioreactor with microchannels. By overlaying multilayer cell sheets on the bioreactors, the vascularized thick tissue could be obtained [55]. In a later work, the same group further transplanted the in vitro prepared vascularized tissue into a rat and successfully realized the blood vessel anastomoses [56]. More recently, Zhang et al. developed a biodegradable scaffold with built-in vasculature. The nanopore and micro-holes within the vessel wall mimic the blood vessel and allow oxygen and nutrient exchange within tissues as thick as 1 mm. The surgical anastomosis to the femoral vessel in a rat allow establishment of immediate blood perfusion [57].

The patch-based strategies have demonstrated dramatical therapeutic effects for hearts with MI. Moreover, one study reports that the integration with host myocardium could be affected by the non-cardiomyocyte epicardium [58]. The ablation of epicardium before transplantation could be challenging. Alternatively, researchers have developed the microneedle-based therapy to break the barrier of the epicardium. The biodegradable microneedle could adhere firmly to the heart while allowing the slow release of encapsulated factors into the myocardium [59]. Moreover, the cells-encapsulated microneedle systems that could continuously release paracrine factors for enhanced heart repair have also been developed [60,61].

## 3. Cell Sheets

In 1990, T. Okano and his group first reported cell sheet technology, using a temperature-responsive cell culture dish that grafted polymers, such as poly-N-isopropylacrylamide (PIPAAm) [62]. In addition, Sekine et al., Wu et al., and Yeh et al. proved that a cell sheet showed superior performance compared with a suspended single cell [63,64,65]. These methods provide a way to use intact cells and secrete ECM to construct 3D structures in sheet form, without the use of a scaffold. This structure mimics in vivo conditions, which means that the cells in the cell sheet have behaviors and functions similar to those of native tissue cells [66].

Skeletal myoblasts (SMs) are the most investigated cell sheets for cardiac repair and have many advantages, including autologous transplantation, ischemia resistance, non-myocyte lineage differentiation, and high proliferative potential [67]. SMs have been used in cardiac regeneration in various animal models [68,69,70,71,72,73,74]. Our group developed a method for the construction of autologous SM cell sheets and reported the first phase I clinical trial on this subject worldwide [75]. The outcomes of a subsequent long-term retrospective study confirmed the recovery of cardiac function and a reduction in mortality [76]. These results support the safety, feasibility, and possible effectiveness in treating end-stage ischemic cardiomyopathy in patients with “no-option”.

MSCs have many characteristics, such as low immunogenicity, paracrine effects, immunosuppression, and tissue repair; thus, these cells have been used in cardiac regeneration for several years [77,78,79]. In MSC transplantation, cell sheet technology eliminates the problems of cell loss and low retention rate. Many studies have reported that bone-marrow-derived mesenchymal stem cell (BM-MSC) sheets and adipose-derived mesenchymal stem cell (AD-MSC) sheets improve cardiac function and the regeneration of cardiac tissue [77,80,81,82]. An increasing number of researchers are focusing on the advantages and clinical application potential of umbilical cord-derived mesenchymal stem cells (UC-MSCs) [83]. Gao et al. fabricated a UC-MSC sheet based on the establishment of a cell bank, and the results showed that UC-MSC sheets could improve cardiac function in a porcine MI model and had no risk of oncogenicity in vivo [84].

MSCs have been widely used to form cell sheets for heart repair in animal models owing to their powerful paracrine ability. However, the effect of paracrine action on heart repair remains limited. The use of cell sheets constructed from iPSC-CMs is considered the most suitable method for heart regeneration treatment. Our group applied iPSC-CM sheets to animal models of MI, including rats and pigs, and the results showed that iPSC-CM sheets could extensively improve cardiac function, such as the left ventricular ejection fraction and fractional shortening, attenuate heart remodeling, diminish the fibrosis rate, and increase neovascularization [85,86,87]. By combining MSCs and hiPSC-CMs, we demonstrated that MSCs could functionally mature hiPSC-CMs, and the mixture could survive and enhance the therapeutic effects for treating MI models [88]. In 2020, our group performed the world’s first clinical trial of iPSC-CM sheet transplantation for severe ischemic cardiomyopathy patients, and so far, the treatment of three patients has been effective. The efficacy was, therefore, confirmed, and no tumorigenesis is detected in the patients [89,90] (Figure 3). In August 2022, our group completed the treatment of the fourth patient at Juntendo University. This was the first transplant performed outside Osaka University. In the future, more participants will be recruited and will receive therapy at multiple sites, including overseas [91].

## 4. Injection of Single Cells and Spheroids

As an easy and straightforward method, injection has long been used to deliver cells into the infarcted myocardium. In patch-based therapies, the epicardium could become a barrier between the CMs in the patch and host CMs, and the cells delivered by injection could help avoid this problem. In a number of reports, the injected cells survived and led to re-muscularization in the infarcted myocardium [92,93,94]. However, graft-related ventricular arrhythmias were also observed, which were due to the spontaneous beating of the injected CMs. In addition, the retention of the injected single cells remains low and requires a large number of cells for injection (~10^9^). Nevertheless, the first-in-man clinical trials based on injecting single hiPSC-CMs have been performed by a team from Nanjing University. Patients receiving the therapy showed markedly improved heart function. However, unsustained irregular heartbeats were noticed [95].

Compared with single-cell suspensions, cardiac spheroids have shown enhanced retention and survival capabilities [96,97,98]. Owing to the auto-assembling properties of cells, scaffold-free spheroids can be generated by using the following methods: (a) culturing on low-attachment plates, (b) hanging drop culturing, (c) microfluidic device generation, (d) bioreactor rotation culturing, and (e) magnetic-based cell coating. Initially, cell spheroid applications focused on drug screening [99] and disease modeling in vitro [100]. The Mummery lab has developed cardiac spheres that contained iPSC-CMs, cardiac fibroblasts, and cardiac endothelial cells. The CMs within these spheres demonstrated structural and functional maturation. Notably, replacing the normal fibroblast with the patient fibroblast would allow for recapitulating arrhythmogenic cardiomyopathy [101].

Human cardiac organoids are used for the modeling of myocardial infarction and drug cardiotoxicity. Until recently, spheroids generated by co-culture of iPSC-CMs, cardiac fibroblasts, and endothelial cells have been used to obtain vascularized cardiac tissues that mimic the properties of the natural human myocardium [102]. These achievements encouraged trials of spheroid transplantation. For instance, Kawaguchi et al. performed spheroid transplantation of iPSC-CMs in rat and swine HF models and confirmed the safety and effectiveness of spheroid transplantation [103]. In addition, Chin et al. described a method of 3D bioprinting of cardiac tissue by using spheroids as “bioinks” and conductive silicon nanowires to provide guidance during positioning. This concept enables precise spatial allocation of spheroids and the creation of more complex cardiac tissue [71] and might serve as a potential method for organ bio-fabrication [104]. In 2020, a clinical trial was initiated by a team at Keio University, which used hiPSC-CM spheres to treat three patients with HF [105].

CardioClusters made of mesenchymal stem cells (MSCs), endothelial progenitor cells (EPCs), and c-Kit+ cardiac interstitial cells (cCICs) have also showed therapeutic effects in response to injury and disease [98] (Figure 4). However, as the presence of transplanted cells in the myocardium is extremely low, more effort is needed to improve the cell survival after transplantation.

To facilitate the even and efficient delivery of spheres into the infarct area, the Keio team also developed a specialized injection device with multiple needles. The device allowed optimal distribution and improved sphere retention in a porcine heart (Figure 4C) [106]. 

In order to facilitate the overall understanding of the tissue engineering technologies mentioned above, we further summarize them in a table (Table 1). 

## 5. Discussion

### 5.1. Challenges and Future Perspectives

#### 5.1.1. Immunological Rejection

Despite the progress in tissue engineering, the clinical translation of hiPSC-CM-based therapies remains challenging. Immunological rejection is the most important issue in hiPSC-CM-based therapies. Although iPSCs could be induced from patient-derived cells for the production of personalized hiPSC-CMs, its cost is high, and its induction is time-consuming. Therefore, allogeneic hiPSC therapy has several advantages. Our group previously found that the immunogenicity of allogeneic iPSC-CMs was reduced by major histocompatibility complex-matched transplantation in a non-human primate model, and a combination of appropriate immune suppression is still required to guarantee engraftment [107]. The mismatch of human leukocyte antigen (HLA) is also a factor for immune rejection, and intensive efforts have been devoted to clustered regularly interspaced short palindromic repeats (CRISPR)/Cas9-based gene editing for HLA molecule disruption [108,109]. Before moving to clinical application, the safety of CRISPR/Cas9 editing, optimal immunosuppressive therapies, and ethical issues must be thoroughly investigated and discussed. Another approach may be the co-transplantation of MSC with hiPSC-CMs. The MSCs have been reported to have profound immunomodulatory properties for their expression of anti-inflammatory cytokines, such as IL-10 and TGFβ [110]. The co-transplantation of iPSC-CM and MSC could reduce the immune rejection [111] and improve the heart function [112] more advantageously than in the iPSC-CM-only group. However, 4 weeks after surgery, only very few transplanted cells survived in the host myocardium, indicating that the immunomodulatory effect of MSC is still limited. Due to this limit, clinical trials are still using immunosuppressive drugs [30,113], which affect the long-term survival of patients. More efforts are needed in the future to investigate the immune regulation for hiPSC-derived therapy.

#### 5.1.2. Cell Retention

In addition to immunological rejection, post-transplantation hypoxia and inflammation can also cause cell loss. These factors make it very difficult to achieve long-term engraftment of transplanted iPSC-derived tissues, especially in large animal models. To help the transplanted tissue survive the harsh environment, Sun et al. developed an improved therapeutic approach by co-transplanting a premade microvessel with hiPSC-CMs, which lead to an increased vessel density, perfusion, cell retention, and functional recovery [114]. However, repeated administration of cardiac progenitor cells has been suggested as a possible solution and has shown marked effectiveness in treating MI [115]. The effectiveness of this method needed to be evaluated with hPSC derived cells, where the injection site may need to be in the myocardium rather than in the LV cavity. In spite of these improvements, the thickness of transplanted tissue tends to be dozens of µm rather than several mm [114]. Due to this limitation, the therapeutic effect of PSC-patch relies more on cytokine factors rather than on remuscularization, often preventing the diseased heart from total recovery. In the future, more efforts are needed to realize transplantation using engineered cardiac tissue with functional blood vessels. The anastomoses with host heart will allow for the immediate perfusion and the retention of the tissue.

#### 5.1.3. Cell Engraftment and Maturation

The functional recovery of the diseased heart requires not only the recruitment of new muscle cells to replace the diseased site but also electrical coupling with the host tissue, and the degree of their maturation is extremely important. As for the electrical coupling between the transplanted tissue and host, the Shiba group has confirmed that the injected CMs can couple well with the host in a non-human primate model [92]. The Eschenhagen group, as well as our group, has also confirmed that the hiPSC-CM patch could form electrical coupling with the host heart in guinea pigs and rat models, respectively [27,116]. Despite the electrical coupling between the graft and the host, there is a number of reports on the injected CMs leading to arrhythmia in the host heart [92,117,118]. This could be due to the immature properties of hPSC-CMs, which include spontaneous beating. On the other hand, the maturation of adult CMs takes years; since the iPSC-CMs in experimental culture only last several weeks, their maturation tends to be weaker. Although it has been reported that the transplanted iPSC-CMs undergo maturation under in vivo conditions [119], many groups including ours have also reported various methods for promoting the maturation of hiPSC-CMs for transplantation [29,88,120,121]. However, these hiPSC-CMs are still less mature than the adult level, which may hamper their therapeutic effects. Future research needs to be focused on further improving the maturation level of hiPSC-CMs in a scalable manner.

#### 5.1.4. Cost

Finally, the cost of iPSC-based therapies is another challenge faced by the patients. To date, autologous iPSC therapies have been proven to be safe in macular degeneration treatment and may be used as rejection-free therapies [122]. In another autologous iPSC-derived cell therapy for Parkinson’s disease, without using immunosuppression, the patient showed positive improvements during the two-year observation [123]. The third autologous iPS-derived therapy is for treating thrombocytopenia. To date, the three doses of autologous iPSC-derived platelets have been performed, and their safety can be confirmed [124]. According to the estimate, the autologous hPSC line that meets the cGMP requirement costs approximately 0.8 million dollars [125,126], and the subsequent scale-up production, as well as the differentiation, would incur an extra cost. The iPSC biobanks established by different countries have reduced costs. In addition, the Masayo Takahashi group showed that automatic manufacturing will dramatically reduce human intervention during large-scale production, which can also significantly reduce production costs [127]. It is necessary to further improve the artificial intelligence of the system and expand its application to the entire hiPSC production process. By implementing the Henry Ford–style production line into hiPSC production, we may bring affordable off-the-shelf hiPSC therapy to patients worldwide.

## 6. Conclusions

We reviewed achievements in cardiac regeneration, including engineered cardiac patches, cell sheets, and cardiac spheroids. Although there are still some issues in cardiac tissue therapy, such as the lack of angiogenesis, inflammation, immune rejection, and maturation, multiple groups around the world have managed to validate the efficacy of hPSC therapy in various animal models. Several clinical trials have been initiated, and exciting progress has been reported, including no signs of tumorigenesis and functional improvement of the heart. Through further technological improvements, therapy based on engineered cardiac tissue could be ready for use by patients with HF in the near future.

## Figures and Tables

**Figure 1 bioengineering-09-00605-f001:**
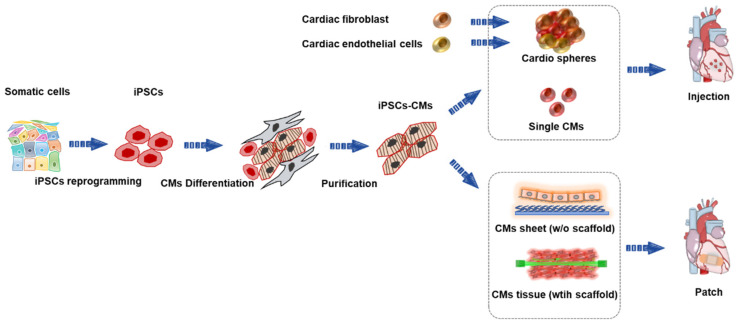
Overview of human induced pluripotent stem-cell-derived cardiomyocytes (iPSC-CMs) for treating heart failure. iPSCs, induced pluripotent stem cells; CMs, cardiomyocytes.

**Figure 2 bioengineering-09-00605-f002:**
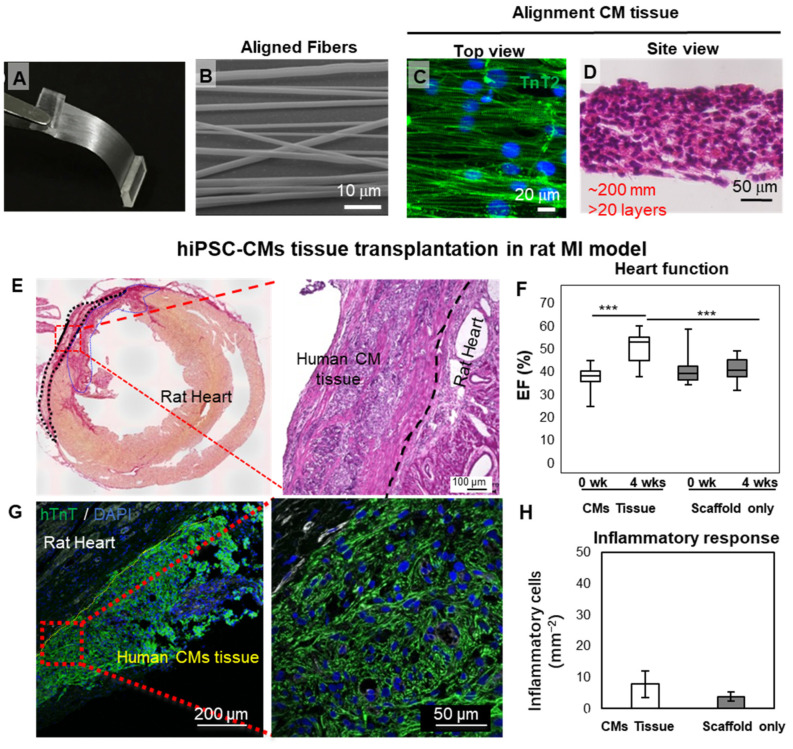
Cardiac-tissue-like constructs formed on aligned nanofibers. (**A**–**D**) Aligned PLGA fiber and the cardiac tissue obtained by seeding hiPSC-CMs. (**E**–**H**) Engraftment of cardiac tissue in an IM rat heart model. A large block of hiPSC-CMs could be found on the host heart four weeks after transplantation. (*** *p*< 0.001) Reprinted from Li et al. [48], with permission. Copyright 2017, the authors. PLGA, poly(lactic-co-glycolic acid); hiPSC-CMs, human induced pluripotent stem cell-derived cardiomyocytes; MI, myocardial infarction; EF, ejection fraction.

**Figure 3 bioengineering-09-00605-f003:**
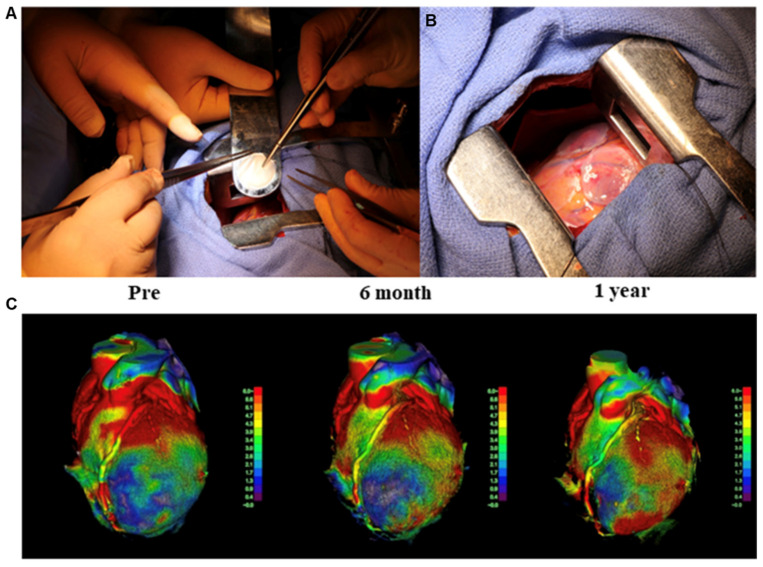
Transplantation of human induced pluripotent stem cell-derived cardiomyocyte (hiPSC-CM) sheet on the heart of a patient, as well as the cardiac movement pattern before and after surgery. (**A**,**B**) First transplantation of hiPSC-CM patches onto the heart surface of a patient with severe cardiomyopathy. (**C**) The moving pattern observed via four-dimensional CT. Reproduced with permission from Miyagawa et al. [89,90]. Copyright 2022, the authors.

**Figure 4 bioengineering-09-00605-f004:**
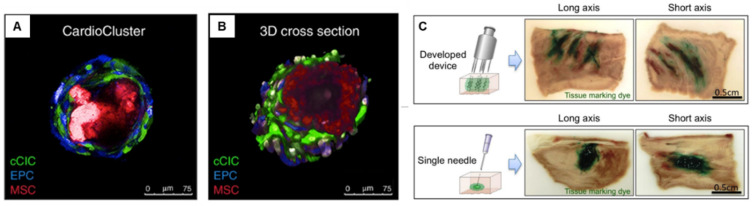
Cardiac sphere used for treating heart failure model. (**A**,**B**) CardioClusters made by mixing cardiac interstitial cells, MSCs, EPCs, and cCICs. Reproduced from Monsanto et al. [98] with permission. Copyright 2020, the Authors. (**C**) Injection device with multiple needles for optimal distribution and improved sphere retention. Reproduced from Tabei et al. [106] with permission. Copyright 2019, the authors. MSCs, mesenchymal stem cells; EPCs, endothelial progenitor cells; cCICs, c-Kit+ cardiac interstitial cells.

**Table 1 bioengineering-09-00605-t001:** Characteristics of tissue-engineering approaches.

Tissue Engineering Approach		Cell Sources	Materials	Species	References
Engineered cardiac tissue	Natural material	hiPSC-CMs & hiPSC-Ecs & hiPSC-SMCs	Fibriogen & thrombin	Swine	[18]
		hMSCs	Collagen/Alginate	Rat	[23]
		MSCs	Hydrogel	Rat	[24]
		hiPSC-CMs	ECM	Rat	[25]
		hiPSC-CMs & hiPSC-ECs	ECM	Guinea pig	[27]
		hiPSC-CMs & hiPSC-Ecs & hiPSC-vascular mural cells	Collagen I & Matrigel	Rat	[28]
		hiPSC-CMs	Fibriogen & thrombin	Guinea pig	[29]
		hESC-cardiovascular progenitors	Fibrin	Human	[30]
		hiPSC-CMs & hiPSC-stromal cells	Bovine collagen type I hydrogel	Human	[31]
		iPSC-CMs & HUVECs	alginate & PEF-Fibrinogen (3D bioprint)	Mice	[37]
	Synthetic material	hiPSC-CMs	PDMS	Guinea pig	[41]
		MSCs	PG	Rat	[46]
		hiPSC-CMs	PLGA	Rat, Porcine	[48,49,50]
		hiPSC-CMs & fibroblasts	Polyglatin 910	Rat	[51]
		Rat CMs & hiPSC-CMs	POMaC	Rat, Porcine	[52]
		Cell free	Carbon nanotube & Bacterial nanocellulose	Canine	[54]
		hESC-CMs & hMSCs & HUVECs	POMaC	Rat	[57]
		MSCs-secreted factors	PLGA & HA	Rat	[59]
Cell sheet		SMs		Rat, Hamster, Canine	[68,69,70,71,72,73,74]
				Human	[75,76]
		BM-MSCs		Porcine	[77]
		ADSCs		Rat, Porcine	[80,81,82]
		UC-MSCs		Mice, Porcine	[84]
		hiPSC-CMs		Rat, Porcine	[85,86,87]
				Human	[89,90]
		hiPSC-CMs & MSCs		Rat	[91]
Injection of single cells		hiPSC-CMs		Monkey, Mice	[92,93]
		hESC-CMs		Monkey	[94]
		hESC-CMs		Human	[95]
Spheroids		hMSCs		Mice	[96]
		MSCs & EPCs & cCICs		Mice	[98]
		hiPSC-CMs		Rat, Swine	[103]
				Human	[105]
				Porcine	[106]

hMSCs, human mesenchymal stem cells; hiPSC, human induced pluripotent stem cell; CMs, cardiomyocytes; SMs, skeletal myoblasts; CSCs, cardiac stem cells; ADSCs, adipose-derived mesenchymal stem cell; BM-MSCs, bone-marrow-derived mesenchymal stem cell; UC-MSCs, umbilical-cord-derived mesenchymal stem cells; ECM, extracellular matrix; PCL, poly(ε-caprolactone); PLGA, poly(lactic-co-glycolic acid).

## Data Availability

Not applicable.

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
