# Peer review of "Engineered Tissue for Cardiac Regeneration: Current Status and Future Perspectives"

_bioengineering, 2022, doi:10.3390/bioengineering9110605_

Round 1

Reviewer 1 Report

Comments and suggestion for authors

1-     The abstract is short and in abstract authors should provide an overview of whole review and mention some the sections ,… with more details

2-     Some of the sections need to refer to proper and relevant references. For instance for the following paragraph and others

“Current HF treatments include prevention and end-of-life care, while surgical heart
transplantation is the only curative remedy. However, the limited supply of donor hearts
severely restricts the use of heart transplantation. Moreover, immunosuppressants, re-
quired following allogeneic transplantation, may cause serious kidney damage or other
problems, such as cancers and infections. Owing to these problems, the emergence of car-
diac tissue engineering has not only provided substantial hope for resolving or rescuing
damaged hearts post-MI but also for providing a permanent treatment strategy.”

3-     Authors should add a section of in vitro tissue constructs for different applications from disease modeling in organ on a chip to drug screening , … ( maybe adding a section called “ cardiac tissue-on-a-chip” can be good

4-     Authors should provide some examples for the following statement and review them: ( two or three examples with more details of the related studies, approaches, applications,…)

“ECP has been widely used in drug screening, disease modeling, and regenerative medicine [10].”

5-     Recently it is found that collagen type V plays a crucial role in heart function after myocardial infarction by affecting on scar size. (https://doi.org/10.1016/j.cell.2020.06.030). It would be good if authors describe the finding of that work and other relevant works for scar tissue and cardiac regeneration after scar formation

6-     Authors can describe more the role of microgrooves and electrical stimulation for cardiac tissue alignment and maturation: here is some relevant references:

https://www.nature.com/articles/nprot.2008.183

https://doi.org/10.1002/smtd.202000438

https://www.ncbi.nlm.nih.gov/pmc/articles/PMC5348721/

7-     3D bioprinting technology also has found great applications in cardiac tissue engineering . it is recommended authors to add a section for application of 3D bioprinting for cardiac tissue engineering

Here are some relevant references and authors can add some relevant figures too

https://doi.org/10.1007/s10856-021-06520-y

https://doi.org/10.1002/adhm.201901794

https://doi.org/10.1016/j.biomaterials.2016.09.003

8-     Recently microneedle-based patches also have been used for stem cell delivery and tissue regeneration applications. Authors can include some applications of microneedles for topic of this review too.

The challenges and future direction section needs further work to have a details of the current challenges and also proposing future works in the field

Reviewer 2 Report

Dear, 

This is an interesting review article. But it should be improved in some parts, please consider below comments:

- Author should mention and explain all natural and synthetic materials as subtitles in "  Natural material scaffold-based approach" and " Synthetic material scaffold-based approach".

-Future direction should be added and explain the advantages and disadvantages and also,  explain the limitations

Reviewer 3 Report

"Engineered tissue for cardiac regeneration: Current status and future perspectives" by Li et al. Is a thorough and engaging review of the current status of tissue engineering.

I scanned the document using the Grammarly plagiarism filter and found no significant plagiarism.  There are, however, quite a few grammar and spelling mistakes throughout the text, which the authors should correct.  Mostly, it is clear and well-written, although I found that section 3, "scaffold-free cell sheets," was sometimes confusing.  In addition, the review tends to favor their own publications rather than give a comprehensive review of the literature.  This is particularly true in the discussion of clinical trials, where they almost exclusively discuss their work.

In addition, I noted a few specific points that I would like to authors to address:

1. The authors need to be clear about discussing human Embryonic Stem Cells and human induced pluripotent stem cells.  It is confusing at points.

2. The authors made a point of only discussing 4-factor induced iPSCs and do not mention 3-factor induced cells.  Can the authors please elaborate on why they think that this is important?

3.  The authors only cite one reference for non-retroviral induction of pluripotent stem cells, and I am sure there are many.

4.  At a few points in the text, the authors imply that pluripotent cells have immune privilege but cite no references to this point. Please discuss this further.

5.  The authors state (line 55-56) that there is"... great expectations for the application of human IPSC-derived CME in regenerative medicine because they have a low cell proliferation capacity."  I think the authors mean they have low tumorigenicity, but they should clarify.

6. At many places throughout the text, the authors ignore the possibility of electrophysiological problems with grafts of cells and sheets; for example, the potential for arrhythmias is not mentioned in the section on skeletal muscle sheets.

7. In figure 1,  they describe cardiosphere as made up of cardiomyocytes, but I believe they are now almost universally made up of cardiomyocyes, fibroblast, and vascular endothelial cells. Indeed, in the text, that is what they say.

8. In figure 2, I believe that the authors meant "side view" rather than "site view."

9.  The authors cite "successful outcomes" from a 2020 clinical trial but do not elaborate; if this is a phase I trial, then "successful outcomes" could be that the procedure was safe.

10. For the following reasons, I believe that it is incorrect to refer to cKit+ cells as cardiac stem cells:

a. The original paper supporting this idea (Kajstura et al.; Circulation 2012 126: 1684) was retracted after several groups failed to reproduce their results; most notably (van Berlo Nature 2015 509:8500). 

b. The two papers they cite to support this idea (Li et al. and Monsanto et al.)  do not address their role as cardiac stem cells.  Indeed Monsanto et al. refer to them as cardiac interstitial cells.  Both papers conclude that the primary function is the induction and maintenance of a stem cell niche by the secretion of ECM.

Beyond these very minor points, I found this to be a very thorough and interesting review of the current state of the art in clinical regenerative medicine.

Round 2

Reviewer 1 Report

The authors addressed my comments. 

 all abbreviations can be double-checked in the text